# Impact of mental health on burden of illness, septicemia and mortality among patients hospitalized for cancer

**Poolakkad S. Satheeshkumar**[1]*, **Roberto Pili**[1], **Sudheer B. Kurunthatil Thazhe**[2], **Rhine Sukumar**[3], **Minu Ponnamma Mohan**[4], **Eric Adjei Boakye**[5], **Joel B. Epstein**[6,7]

**1** Department of Medicine, Division of Hematology and Oncology, University at Buffalo, Buffalo, New York, United States of America, **2** WWRC, Hamad Medical Corporation, Doha, Qatar, **3** Naseem Al Rabeeh Medical Center, Doha, Qatar, **4** Boston Medical Center, Boston University School of Medicine, Boston, Massachusetts, United States of America, **5** Department of Public Health Sciences, Department of Otolaryngology-Head & Neck Surgery, Michigan State University College of Human Medicine & Henry Ford Health, Detroit, Michigan, United States of America, **6** City of Hope Comprehensive Cancer Center, Duarte, CA, United States of America, **7** Cedars-Sinai Health System, Los Angeles, California, United States of America

* spoolakk@buffalo.edu

**Data Availability Statement:** We used data from the Publicly available United States' 2017 National Inpatient Sample database obtained from

## Abstract

Mental health problems are increasingly relevant for cancer patients struggling with the disease and its treatment. The purpose of this study was to further characterize and contrast variances between Mental illness (MI)—cognitive disorders—and clinical outcomes in patients hospitalized in the United States in 2017 for the treatment of prostate (PC), lung (LC), leukemia, and oral cavity, lip, and pharyngeal cancers (OPC). While accounting for patient and clinical characteristics, we used generalized linear models to evaluate the association between MI and outcomes—mortality, septicemia, weight loss, fluid and electrolyte imbalance, and illness burden (length of stay (LOS) and total charges). There were 16,910 (Weighted, original numbers) patients with MI among 209,410 PC patients. In the adjusted analysis, PC patients with MI had a prolonged LOS, coefficient: 1.52; 1.41–1.64. In addition, MI were associated with increased odds of septicemia (1.36; 1.22–1.51), weight loss (1.38; 1.23–1.56), and fluid and electrolyte imbalance (1.33; 1.21–1.53). These findings were comparable for the lung, leukemia, and oral cavity, lip, and pharyngeal cancers. In addition, unlike other cancer cohorts, MI were associated with increased odds of in-hospital mortality in PC patients, 1.42, 1.21–1.58. Patients diagnosed with cancer who also suffered from cognitive impairments had poor clinical outcomes. The findings of this study bring to light a gap in the existing literature on cancer, and the recommendations emphasize the significance of psychosocial support in reaching a more favorable prognosis and improving quality of life.

## Introduction

Patients undergoing cancer treatment face an increased vulnerability to negative mental health outcomes, particularly those in advanced stages of the disease. These individuals are more

Healthcare Cost and Utilization Project (HCUP) of the Agency for Healthcare Research and Quality (AHRQ). The data is a publicly available anonymous largest all-payer inpatient healthcare database in the US that is accessible to the general public. Due to these user agreements, we are prohibited from sharing the data with any repository or publication. Conversely, the information is accessible to any anyone from the general public who has an interest in it. If the journal requires to contact the data distributor, please email: HCUP-RequestData@AHRQ.gov.

**Funding:** The authors received no specific funding for this work.

**Competing interests:** The authors have declared that no competing interests exist.

susceptible to acute triggers such as infection or the poor consequences of cancer therapy, which can further worsen their mental well-being [1–3]. Furthermore, advanced-stage, old age, or dementia that already exists or is getting worse are added to the complexity [4, 5]. Health care providers face difficulties in screening, diagnosing, and monitoring MI such as cognitive disorders in cancer care, which cause stress for patients and their caregivers and presents obstacles for medical staff [6–8]. More than one in five U.S. adults experienced MI in 2021, totaling 57.8 million people, and one in two may receive a diagnosis at some point in their lives [9–11]. A wide range of disorders, from mild to severe, fall under the umbrella of "mental illness." Up to one-third of the patients undergoing cancer treatment in hospitals are diagnosed with a prevalent mental health problem [11, 12]. Up to three times as many people in this community suffer from serious mental illnesses than they do in the overall population [13, 14]. Moreover, there is an enormous gap in knowledge when it comes to the impact of the mental health conditions on cancer patients.

The published literature focuses on out-patients, with only a few studies including in-patient cohorts. These studies assessed various interventions, such as massage [15], cognitive-behavioral therapy (CBT) [16], collaborative care [17], cost effectiveness [18], coping skills [19], and homebased walking [20]. The outcomes encompassed sleep quality and exhaustion [20], pain and insomnia [21], physical function [22], psychological factors [23], cost [24], and others [25–29]. Nevertheless, research investigating in-hospital outcomes, including mortality, burden of illness, and hospital-acquired complications, is a critical component of patient care that must be taken into account when assessing quality of life. Furthermore, there is a paucity of research examining in-hospital complications on cancer patients who are admitted to hospitals for treatment, indicating lack of research on the outcomes of hospitalized cancer patients with mental illness, resulting in a gap in our present understanding of the burden of illness, problems associated with hospitalization, and mortality rates. The link between these factors have been ascribed to discrepancies in healthcare accessibility encountered by individuals with mental illness. Individuals suffering from psychiatric illness frequently encounter restricted availability of healthcare services, delayed identification of their condition, decreased compliance with treatment, and poor health seeking behaviors. Thus, we hypothesize an association between cancer patients who also suffer from MI and unfavorable in-hospital outcomes, including but not limited to burden of illness, in-hospital mortality, and complications associated with the hospital setting.

This study focuses the risks and outcomes of MI, specifically cognitive disorders, among cancer patients. We aimed to test this hypothesis by analyzing various outcomes, including the burden of illness (such as length of stay and total charge), weight loss, septicemia, fluid and electrolyte imbalance, and in-hospital mortality among cancer patients who were admitted for hospitalization. To conduct this investigation, we utilized the database of the United States' National Inpatient Sample and analyzed patients who were hospitalized for the treatment of leukemia, PC, LC, and OPC. The significance of this study lies in providing valuable information about the impact of current patient care strategies and the impact of mental health on cancer patients.

## Methods

### Data source and study population

The 2017 National Inpatient Sample (NIS) database from the Healthcare Cost and Utilization Project (HCUP) at the Agency for Healthcare Research and Quality was used for this analysis [30].

## Ethics statement

The HCUP does not require that users receive IRB review from their institutions, hence has not submitted for the Institutional Review Board (IRB) because it used publicly available data that had been de-identified or contain limited data set. Hence, a patient consent is not possible. Further this study included only adults >18 years age, and hence study doesn't included children under 18 years of age.

Patient demographics, medical history, outcomes during hospital stays, hospital details, and costs are all included in the NIS dataset. This study included patients who were hospitalized for the treatment of leukemia, prostate (PC), lung cancer (LC), and cancers of the oral cavity, lip, and pharynx (OPC). Cancer types (prostate (PC), lung (LC), and cancers of the oral cavity, lip, and pharynx (OPC), and leukemia were identified using ICD-10-CM billing codes (S1 File). We plan to incorporate multiple cancer cohorts admitted for treatment of liquid and solid cancers. This approach allows us to comprehensively analyze the validation across different cohorts, despite their differences in cancer types. Specifically, we aim to examine the impact of MI, on patients with prostate cancer (PC), lung cancer (LC), oropharyngeal cancer (OPC), and leukemia. We expect to observe a consistent influence of these mental illnesses on the mentioned cancer types. In order to confirm the accuracy of our results in cases where consistent cancer treatment approaches were used or a specific group of patients were identified, we specifically selected leukemia patients who underwent allogenic hematopoietic stem cell transplant, as well as LC patients who received lobectomy treatment.

## Measures

The primary independent variables included MI (delirium, dementia, and amnestic and other cognitive disorders (cognitive disorders)) among patients with PC, LC, OPC, and leukemia. These were identified using ICD-10-CM billing codes (S1 File).

The outcome variables included burden of illness (in-hospital length of stay (LOS), and total charges), in-hospital mortality, septicemia, weight loss, and fluid and electrolyte imbalance. The term "total charges" is used to describe the complete cost of care/total costs incurred during a hospital stay. Typically, professional fees and non-covered expenditures are excluded from total charges. During HCUP processing, total charges that include professional fees are deducted, and the total charge was then adjusted for the current year. Patients are deemed to have died "in hospital" if they are reported as "alive" or "dead" at the time of release. We log-transformed total charges and LOS and presented the geometric mean due to the non-normal distribution. To avoid a negative log, LOS of 0 days was imputed with a value of 0.0001. Septicemia, weight loss, and fluid and electrolyte imbalance were defined based on ICD-10-CM billing codes (S1 File).

Patient and clinical factors were included as covariates. Age, gender, primary payer (Medicare, Medicaid, Private Insurance, and Others), median household income by zip code (first to fourth quartile), and urban/rural status (using a six-category urban-rural classification scheme for United States counties developed by the National Center for Health Statistics) were among the patient characteristics. Other covariates included place of origin (transferred-in vs. not transferred), the nature of their hospitalization (elective vs. non-elective), and their comorbidity status, as evaluated by the Elixhauser comorbidity index [31].

## Statistical analysis

Descriptive statistics were used to describe the demographic and clinical characteristics of each cancer cohort. In addition, we used survey-weighted generalized linear models [32] (Svyglm) to investigate the association between exposure status MI and the outcomes of burden of illness

(in-hospital length of stay (LOS), and total charges), in-hospital mortality, septicemia, weight loss, and fluid and electrolyte imbalance for each cancer cohort. The association between key independent variables and outcome in the two patient populations was examined using univariate and multivariate analyses. We adjusted the multivariate models for patient age, gender, payer type, patient location, race, and median household income, as well as co-morbidities score, hospital discharge status, and median household income. When fitting the Svyglm to the models with dichotomous outcomes, we used a quasibinomial with a family reference. In all of our experiments, we used two-tailed probability distributions and set the threshold for statistical significance at $P < 0.05$. All statistical analyses were run in the R 3.6.3 statistical computing environment from the R Foundation for Statistical Computing in Vienna, Austria.

## Results

### Prostate cancers

Out of the total of 209,410 PCs, there were 16,910 patients who were diagnosed with MI. The median age of patients with PC was 70 years, with an interquartile range (IQR) of 63 to 78. Among PC patients with MI, the median age was 84 years, with an IQR of 78 to 88. In the unadjusted analysis, patients diagnosed with MI showed an extended length of stay (LOS), a greater rate of mortality during their hospital stay, weight loss, imbalances in fluid and electrolyte levels, and septicemia (Table 1). All PC case counts, rates, and estimates are calculated using weighted data.

The adjusted analysis showed that PC patients with MI were associated with longer LOS (coefficient [coef] = 1.52; 95% confidence interval [CI]: 1.41–1.64). In addition, patients with MI were associated with a greater likelihood of developing septicemia (adjusted odds ratio (aOR):1.36; 95% CI: 1.22–1.51), weight loss (aOR:1.38; 95% CI: 1.23–1.56), fluid and electrolyte imbalance (aOR: 1.33; 95% CI: 1.21–1.46), and in-hospital mortality (aOR:1.42; 95% CI: 1.21–1.68) Table 2.

### Lung cancers

The study included a total of 403,010 LC cohorts. Among this group, 21,895 patients were diagnosed with MI. The median age of patients with LC was 69 years with an IQR of 61 to 76. Among LC patients with MI, the median age was 78 years with an IQR of 72 to 84. In the unadjusted analysis, LC patients with MI showed extended length of stay (LOS), increased total charges, higher in-hospital mortality rate, and a greater incidence of septicemia in the unadjusted analysis (Table 3). All LC case counts, rates, and estimates are calculated using weighted data.

In the adjusted analysis, LC patients with MI were associated with a longer length of stay (coef = 1.36, 95%CI: 1.28–1.45) and higher total charges (coef = 1.10, 95%CI: 1.05–1.15). Additionally, LC patients with MI in the LC group were associated with an increased likelihood of developing septicemia (aOR 1.18; 95% CI: 1.08–1.29), weight loss (aOR 1.53; 95% CI: 1.29–1.53), and fluid and electrolyte imbalance (aOR: 1.16; 95% CI: 1.08–1.25) Table 2.

### OPC

In the study there were 54,264 total number of OPC, among them were 2,275 patients with MI. The median age of the patients with OPC was 63.00 [56.00, 72.00], while among those with OPC and MI, it was 77.00 [67.00, 84.25]. In the unadjusted analysis, OPC patients diagnosed with MI exhibited increased length of stay (LOS), higher total charges, higher in-hospital mortality rate, and a greater incidence of septicemia (Table 4). All OPC case counts, rates, and estimates are calculated using weighted data.

**Table 1. Baseline characteristics of prostate cancer patients with and without MI.**

| | Prostate cancer without MI (Weighted) | Prostate cancer with MI (Weighted) | P- value |
|---|---|---|---|
| **N** | 192500 | 16910 | |
| **AGE (median [IQR])** | 70 [63, 78] | 84 [78, 88] | <0.001 |
| **RACE (%)** | | | <0.001 |
| White | 129720 (70.1) | 10895 (66.1) | |
| Black | 32230 (17.4) | 3460 (21.0) | |
| Hispanic | 12995 (7.0) | 1275 (7.7) | |
| Others | 10180 (5.5) | 845 (5.1) | |
| **Expected primary payer (%)** | | | <0.001 |
| Medicare | 122419 (63.7) | 15220 (90.1) | |
| Medicaid | 907 (4.7) | 355 (2.1) | |
| Private insurance | 53220 (27.7) | 905 (5.4) | |
| Self-pay, No charge and other | 7475 (3.9) | 405 (2.4) | |
| **Median household income (based on current year)** | | | 0.001 |
| 0-25th percentile | 113800 (30.3) | 6535 (30.3) | |
| 26th to 50th percentile | 103010 (27.4) | 5600 (25.9) | |
| 51st to 75th percentile | 87325 (23.2) | 4795 (22.2) | |
| 76th to 100th percentile | 71600 (19.1) | 4670 (21.6) | |
| **Patient Location: NCHS Urban-Rural Code (%)** | | | <0.001 |
| "Central" counties of metro areas of > = 1 million population | 56185 (29.3) | 5895 (34.9) | |
| "Fringe" counties of metro areas of > = 1 million population | 49100 (25.6) | 4205 (24.9) | |
| Counties in metro areas of 250,000–999,999 population. | 37605 (19.6) | 2815 (16.7) | |
| Counties in metro areas of 50,000–249,999 population | 17815 (9.3) | 1475 (8.7) | |
| Micropolitan counties & Not metropolitan or micropolitan counties. | 31090 (16.2) | 2485 (14.7) | |
| **Admission type (%)** | | | <0.001 |
| Elective | 82115 (42.7) | 1295 (7.7) | |
| **Indicator of a transfer out of the hospital** | | | <0.001 |
| Transferred out | 30855 (16.0) | 8655 (51.3) | |
| **Weighted Elixir score mean (SD))** | 13.9 (10.9) | 19.9 (10.2) | <0.001 |
| **Length of Stay (Geometric mean)** | 2.4 days | 4.5 days | <0.001 |
| **Total Charge (Geometric mean)** | $41764 | $38166 | 0.01 |
| **Septicemia (%)** | 18255 (9.5) | 3055 (18.1) | <0.001 |
| **Weight loss (%)** | 17500 (9.1) | 3020 (17.9) | <0.001 |
| **Fluid and electrolyte imbalance (%)** | 50545 (26.3) | 7765 (45.9) | <0.001 |
| **In-hospital Mortality (%)** | 5720 (3.0) | 1180 (7.0) | <0.001 |
| **Age group (%)** | | | <0.001 |
| 45–54 years | 10120.0 (5.5) | 25 (0.1) | |
| 55–64 years | 44595.0 (24.4) | 365 (2.2) | |
| >65 years | 127694.9 (70.0) | 16415 (97.7) | |
| **LOS group (%)** | | | <0.001 |
| 1–9 | 116150 (89.1) | 12375 (81.7) | |
| 10–19 | 11375 (8.7) | 2095 (13.8) | |
| 20–29 | 2035 (1.6) | 430 (2.8) | |
| >30 | 825 (0.6) | 245 (1.6) | |

Abbreviations: SD, Standard deviation; NCHS, National Center for Health Statistics; $, United States' Dollar. LOS: Length of Stay.

Note: All frequencies and percentages are weighted

**Table 2. Multivariate analysis of cancer patients with MI.**

| Outcome | Lung cancer patients with delirium, dementia, and amnestic and other MI (Coefficients, 95% CI, and P value). Adjusted Analysis | Prostate cancer patients with delirium, dementia, and amnestic and other MI (Coefficients, 95%CI, and P value). Adjusted Analysis | cancers of the lip, oral cavity and pharynx patients with delirium, dementia, and amnestic and other MI (Coefficients, 95%CI, and P value). Adjusted Analysis | Leukemia patients with delirium, dementia, and amnestic and other MI (Coefficients, 95%CI, and P value). Adjusted Analysis |
|---|---|---|---|---|
| LOS | 1.36; 95% CI = 1.28–1.45: p < 0.001 | 1.52, 95%CI: 1.41–1.64: p < 0.001 | 1.87, 95%CI: 1.56–2.24: p < 0.001 | 1.24, 95%CI: 1.03–1.51: p = 0.02 |
| Total charges | 1.10, 95% CI = 1.05–1.15; p<0.001 | 1.03, 95% CI = 0.98–1.09; p = 0.23 | 1.35, 95% CI = 1.17–1.54; p<0.001 | 0.96, 95% CI = 0.82–1.11; p = 0.56 |
| Septicemia | aOR = 1.18; 95% CI = 1.08–1.29, p = 0.0002 | aOR = 1.36; 95% CI = 1.22–1.51, p = 0.0002 | aOR = 1.39; 95% CI = 1.07–1.80, p = 0.01 | aOR = 1.32; 95% CI = 1.07–1.63; p = 0.009 |
| Weight loss | aOR = 1.41; 95% CI = 1.29–1.53, p<0.001 | aOR = 1.38; 95% CI = 1.23–1.56, p<0.001 | aOR = 0.92; 95% CI = 0.72–1.17, p = 0.48 | aOR = 1.34; 95% CI = 1.05–1.69; p = 0.02 |
| FED | aOR = 1.16; 95% CI = 1.08–1.25, p<0.001 | aOR = 1.33; 95% CI = 1.21–1.46, p<0.001 | aOR = 1.38; 95% CI = 1.10–1.72 p = 0.006 | aOR = 1.20; 95% CI = 0.99–1.47; p = 0.06 |
| Mortality | aOR = 1.11; 95% CI = 0.99–1.25, p = 0.08 | aOR = 1.42; 95% CI = 1.21–1.68, p<0.001 | aOR = 0.85; 95% CI = 0.56–1.28, p = 0.44 | aOR = 1.14; 95% CI = 0.85–1.54; p = 0.39 |

Length of stay, LOS; Fluid and electrolyte imbalance, FED

In the adjusted analysis, patients with MI in the OPC group were associated with longer length of stay (coef = 1.87, 95% CI: 1.56–2.24), higher total charges (coef = 1.35, 95% CI: 1.17–1.54), and associated with increased odds of developing septicemia (aOR = 1.39, 95% CI: 1.07–1.80) as well as fluid and electrolyte imbalance (aOR = 1.38, 95% CI: 1.10–1.72) Table 2.

### Leukemia

Out of a total of 71,779 patients diagnosed with leukemia, 3050 were diagnosed with MI. The median age of leukemia patients was 66.00 and IQR of 53.00 and 75.00. Among leukemia patients with MI, the median age was 81.00 and IQR of 73.00, 87.00. In the unadjusted analysis, leukemia patients with MI were found to have a extended LOS, higher rates of in-hospital mortality, and increased incidence of septicemia (Table 5). All leukemia case counts, rates, and estimates are calculated using weighted data.

In the adjusted analysis, leukemia patients who had MI were associated with longer LOS (coef = 1.24, 95% CI: 1.03–1.51). Further, leukemia patients who experience MI were associated with a greater likelihood of developing septicemia (1.32, 1.07–1.63) and experiencing weight loss (1.34; 1.05–1.69) (Table 2).

### Sub-group analysis

To further, validate our findings where uniform cancer management strategies were adopted or a unique patient sub-population were diagnosed, we chose leukemia patients receiving allogenic hematopoietic stem cell transplant, and LC patients receiving lobectomy treatment. Thus, in the sub-group, we stratified our analysis to those received allogenic hematopoietic stem cell transplant as part of the cancer treatment, and among those treated for cancer of the lung (*Malignant neoplasm of LC*, Resection of Lung Lobe, Open Approach).

### Allogenic hematopoietic stem cell transplant (HSCT)

There were 6695 (Weighted) Allogenic hematopoietic stem cell transplant patients, and among them there were 135 patients with MI. Median [IQR] age of the HSCT patients were 56 [41, 64], and this was 61 [56, 64.25] among patients with MI.

**Table 3. Baseline characteristics of lung cancer patients with and without MI.**

| | Lung cancer without MI (Weighted) | Lung cancer with MI (Weighted) | P- value |
|---|---|---|---|
| **N** | 381115 | 21895 | |
| **AGE (median [IQR])** | 69 [61, 76] | 78.00 [72, 84] | <0.001 |
| **FEMALE** | | | <0.001 |
| | 185440 (48.7) | 11490 (52.5) | |
| **RACE (%)** | | | 0.020 |
| White | 287785 (77.6) | 16235 (76.1) | |
| Black | 45535 (12.3) | 2985 (14.0) | |
| Hispanic | 16960 (4.6) | 1010 (4.7) | |
| Others | 20480 (5.5) | 1110 (5.2) | |
| **Expected primary payer (%)** | | | <0.001 |
| Medicare | 249180 (65.5) | 18650 (85.3) | |
| Medicaid | 38925.0 (10.2) | 880 (4.0) | |
| Private insurance | 75770.0 (19.9) | 1815 (8.3) | |
| Self-pay, No charge and other | 16735.0 (4.4) | 525 (2.4) | |
| **Median household income (based on current year)** | | | 0.001 |
| 0-25th percentile | 113800 (30.3) | 6535 (30.3) | |
| 2 6th to 50th percentile | 103010 (27.4) | 5600 (25.9) | |
| 51st to 75th percentile | 87325 (23.2) | 4795 (22.2) | |
| 76th to 100th percentile | 71600 (19.1) | 4670 (21.6) | |
| **Patient Location: NCHS Urban-Rural Code (%)** | | | <0.001 |
| "Central" counties of metro areas of > = 1 million population | 97555.0 (25.7) | 6485.0 (29.7) | |
| "Fringe" counties of metro areas of > = 1 million population | 96450.0 (25.4) | 5585.0 (25.6) | |
| Counties in metro areas of 250,000–999,999 population. | 76735.0 (20.2) | 4150.0 (19.0) | |
| Counties in metro areas of 50,000–249,999 population | 38125.0 (10.0) | 2105.0 (9.6) | |
| Micropolitan counties & Not metropolitan or micropolitan counties. | 71415.0 (18.8) | 3510.0 (16.1) | |
| **Admission type (%)** | | | <0.001 |
| Elective | 72810 (19.1) | 2410 (11.0) | |
| **Indicator of a transfer out of the hospital** | | | <0.001 |
| Transferred out | 77010 (20.2) | 9835 (45.0) | |
| **Weighted Elixir score mean (SD))** | 19.00 (9.96) | 20.87 (9.98) | <0.001 |
| **Length of Stay (Geometric mean)** | 3.7 days | 4.8 days | <0.001 |
| **Total Charge (Geometric mean)** | $41764 | $43538 | 0.01 |
| **Septicemia (%)** | 47350 (12.4) | 3300 (15.1) | <0.001 |
| **Weight loss (%)** | 78330 (20.6) | 6205.0 (28.3) | <0.001 |
| **Fluid and electrolyte imbalance (%)** | 150940 (39.6) | 9990 (45.6) | <0.001 |
| **In-hospital Mortality (%)** | 32750 (8.6) | 2320 (10.6) | <0.001 |
| **Age group (%)** | | | <0.001 |
| **45–54 years** | 27015.0 (7.6) | 305.0 (1.4) | |
| **55–64 years** | 91665.0 (25.7) | 1705.0 (8.0) | |
| **>65 years** | 238520.0 (66.8) | 19425.0 (90.6) | |
| **LOS group (%)** | | | <0.001 |
| **1–9** | 281750 (85.4) | 15050 (78.6) | |
| **10–19** | 38900 (11.8) | 3085 (16.1) | |
| **20–29** | 6520 (2.0) | 605 (3.2) | |
| **>30** | 2800.0 (0.8) | 400 (2.1) | |

Abbreviations: SD, Standard deviation; NCHS, National Center for Health Statistics; $, United States' Dollar. LOS: Length of Stay.

Note: All frequencies and percentages are weighted

**Table 4. Baseline characteristics of cancers of the lip, oral cavity and pharynx patients with and without MI.**

| | Cancers of the lip, oral cavity and Pharynx without MI (Weighted) | Cancers of the lip, oral cavity and Pharynx with MI (Weighted) | P- value |
|---|---|---|---|
| **N** | 51990 | 2275 | |
| **AGE (median [IQR])** | 63 [56, 72] | 77 [67, 84.3] | <0.001 |
| **Female** | | | 0.01 |
| | 14810 (28.5) | 775 (34.1) | |
| **RACE (%)** | | | 0.051 |
| White | 37585 (74.9) | 1625 (73.0) | |
| Black | 5265 (10.5) | 305 (13.7) | |
| Hispanic | 3230 (6.4) | 160 (7.2) | |
| Others | 4105 (8.2) | 135 (6.1) | |
| **Expected primary payer (%)** | | | <0.001 |
| Medicare | 25420 (49.0) | 1860.0 (81.9) | |
| Medicaid | 8170 (15.7) | 180.0 (7.9) | |
| Private insurance | 15400 (29.7) | 165.0 (7.3) | |
| Self-pay, No charge and other | 2910 (5.6) | 65.0 (2.9) | |
| **Median household income (based on current year)** | | | 0.23 |
| 0-25th percentile | 14510 (28.5) | 595 (26.3) | |
| 26th to 50th percentile | 13350 (26.2) | 690 (30.5) | |
| 51st to 75th percentile | 12285 (24.1) | 500 (22.1) | |
| 76th to 100th percentile | 10830 (21.2) | 480 (21.2) | |
| **Patient Location: NCHS Urban-Rural Code (%)** | | | 0.886 |
| "Central" counties of metro areas of > = 1 million population | 14925 (28.8) | 675 (29.7) | |
| "Fringe" counties of metro areas of > = 1 million population | 13290 (25.7) | 600.0 (26.4) | |
| Counties in metro areas of 250,000–999,999 population. | 10585 (20.4) | 470.0 (20.7) | |
| Counties in metro areas of 50,000–249,999 population | 4965 (9.6) | 215.0 (9.5) | |
| Micropolitan counties & Not metropolitan or micropolitan counties. | 8020 (15.5) | 310.0 (13.7) | |
| **Admission type (%)** | | | <0.001 |
| Elective | 19125 (36.9) | 520 (22.9) | |
| **Indicator of a transfer out of the hospital** | | | <0.001 |
| Transferred out | 9320 (17.9) | 1140 (50.1) | |
| **Weighted Elixir score mean (SD))** | 15.52 (9.39) | 19.84 (10.20) | <0.001 |
| **Length of Stay (Geometric mean)** | 3.6 days | 6.02 days | <0.001 |
| **Total Charge (Geometric mean)** | $48644 | $58251 | <0.001 |
| **Septicemia (%)** | 5510.0 (10.6) | 395 (17.4) | <0.001 |
| **Weight loss (%)** | 7765 (45.9) | 835 (36.7) | <0.001 |
| **Fluid and electrolyte imbalance** | 20210 (38.9) | 1205 (53.0) | <0.001 |
| **In-hospital Mortality (%)** | 2220.0 (4.3) | 140 (6.2) | 0.05 |
| **Age group (%)** | | | <0.001 |
| 45–54 years | 7765 (16.9) | 65 (3.0) | |
| 55–64 years | 15840 (34.5) | 295 (13.7) | |
| >65 years | 22325 (48.6) | 1800 (83.3) | |
| **LOS group (%)** | | | <0.001 |
| 1–9 | 36060 (83.0) | 1385.(67.6) | |

*(Continued)*

**Table 4.** (Continued)

|  | Cancers of the lip, oral cavity and Pharynx without MI (Weighted) | Cancers of the lip, oral cavity and Pharynx with MI (Weighted) | P- value |
|---|---|---|---|
| 10–19 | 5510 (12.7) | 435 (21.2) |  |
| 20–29 | 1160 (2.7) | 130 (6.3) |  |
| >30 | 720 (1.7) | 100 (4.9) |  |

Abbreviations: SD, Standard deviation; NCHS, National Center for Health Statistics; $, United States' Dollar. LOS: Length of Stay.

Note: All frequencies and percentages are weighted

In the unadjusted analysis, HSCT patients with MI were reported to have a higher total charge ($, median [IQR], 635374.38 [430043.56, 950720.19] vs. 405590 [273227.50, 589139.57]; longer LOS (median [IQR], 33 [26.50, 47.50] vs. 26 [22, 31.00]; higher in-hospital mortality (22% vs. 4%); higher septicemia (44.4% vs. 11.6%); higher fluid and electrolyte imbalance (85.2% vs. 48.7%).

In the adjusted analysis, patients with MI were associated with longer LOS, coef: 1.65, 95% CI: 1.26–2.18, and higher total charges, 2.19; 1.36–3.51. Further HSCT patients with MI were associated with septicemia, 4.63, 1.86–11.57, and fluid and electrolyte imbalance, 5.02; 1.15–21.9. However, HSCT patients with MI were not associated with in-hospital mortality, 3.06, 0.82–11.46.

## Malignant neoplasm of LC, resection of lung lobe, open approach

There were 8295 (Weighted) malignant neoplasms of lung cancer lobectomy patients, and among them there were 195 patients with MI. Median [IQR] age of the malignant neoplasms of LC, lobectomy patients were 67 [59, 73], and this was 72 [69, 77] among patients with MI.

In the unadjusted analysis, malignant neoplasms of lunger cancer lobectomy patients with MI reported longer LOS (median [IQR], 9 [6, 15] vs. 6 [4, 9], weight loss (20.8% vs. 6.4%), fluid and electrolyte imbalance (41.7% vs. 21.9%), higher mortality (12.5% vs. 2.6%), and septicemia (10.4% vs. 3.7%).

In the adjusted analysis, patients with MI were associated with longer LOS, coefficient: 1.86, 95%CI: 1.33–2.62, and higher total charges: 1.56, 95%CI: 1.06–2.29.

## Discussion

We found that cancer patients with MI has increased LOS, total charges, septicemia, weight loss, and fluid and electrolyte imbalance. Although cancer patients have been conjectured in previous studies concerning the dismal prognosis of mental illnesses—MI—, this is the first time it has been reported from a large examination of real-world data from an in-patient hospital record [15–29]. The Diagnostic and Statistical Manual of Mental Disorders (DSM) MI—delirium, dementia, amnesia, and other MI—were used to categorize mental illnesses (MI) in our study [33]. The broad spectrum of cognitive diseases is made up of a diverse group of delirium, dementia, amnesia, and other MI where altered cognition caused by recognized disease entities is the fundamental distinguishing feature [34]. Recently, the MI were named as neuro MI with conceptualization of *DSM-V* [35, 36]. These classifications are intended to promote rigorous clinical reasoning, and the DSM changes are intended to enhance differential diagnostic consideration [37]. As a result, we believe that incorporating delirium, dementia, amnesia, and other cognitive impairments as the illness category could be a useful tool for gauging the cognitive disorder outcome among cancer patients.

**Table 5. Baseline characteristics of leukemia patients with and without MI.**

| | Leukemia patients without MI (Weighted) | Leukemia patients with MI (Weighted) | P- value |
|---|---|---|---|
| **N** | 68730 | 3050 | |
| **AGE (median [IQR])** | 66 [53, 75] | 77 [67, 84.3] | <0.001 |
| **Female** | | | 0.03 |
| | 30125 (43.8) | 1490 (48.9) | |
| **RACE (%)** | | | 0.120 |
| White | 47920 (72.5) | 2205 (74.2) | |
| Black | 7050 (10.7) | 380 (12.8) | |
| Hispanic | 5720 (8.7) | 200 (6.7) | |
| Others | 5370 (8.1) | 185 (6.2) | |
| **Expected primary payer (%)** | | | <0.001 |
| Medicare | 37630 (54.9) | 2650.0 (86.9) | |
| Medicaid | 7660 (11.2) | 125.0 (4.1) | |
| Private insurance | 20240 (29.5) | 225.0 (7.4) | |
| Self-pay, No charge and other | 3050 (4.4) | 50.0 (1.6) | |
| **Median household income (based on current year)** | | | 0.27 |
| 0-25th percentile | 16160 (23.9) | 770 (25.5) | |
| 26th to 50th percentile | 17875 (26.4) | 730 (24.2) | |
| 51st to 75th percentile | 17145 (25.4) | 700 (23.2) | |
| 76th to 100th percentile | 16440 (24.3) | 815 (27.0) | |
| **Patient Location: NCHS Urban-Rural Code (%)** | | | 0.07 |
| "Central" counties of metro areas of > = 1 million population | 20320 (29.7) | 1005 (33.0) | |
| "Fringe" counties of metro areas of > = 1 million population | 18165 (26.5) | 810 (26.6) | |
| Counties in metro areas of 250,000–999,999 population. | 13065 (19.1) | 635 (20.9) | |
| Counties in metro areas of 50,000–249,999 population | 6260 (9.1) | 250 (8.2) | |
| Micropolitan counties & Not metropolitan or micropolitan counties. | 10635 (15.5) | 345 (11.3) | |
| **Admission type (%)** | | | <0.001 |
| Elective | 14045 (20.5) | 240 (7.9) | |
| **Indicator of a transfer out of the hospital** | | | <0.001 |
| Transferred out | 9320 (17.9) | 1140 (50.1) | |
| **Weighted Elixir score mean (SD))** | 8.57 (8.62) | 13.34 (8.61) | <0.001 |
| **Length of Stay (Geometric mean)** | 5.03 days | 5.24 days | <0.001 |
| **Total Charge (Geometric mean)** | $ 57449 | $ 49667 | <0.001 |
| **Septicemia (%)** | 11810 (17.2) | 730 (23.9) | <0.001 |
| **Weight loss (%)** | 9750 (14.2) | 675 (22.1) | <0.001 |
| **Fluid and electrolyte imbalance (%)** | 26505 (38.6) | 1540 (50.5) | <0.001 |
| **In-hospital Mortality (%)** | 4875 (7.1) | 350 (11.5) | <0.001 |
| **Age group (%)** | | | <0.001 |
| 45–54 years | 7935 (14.2) | 50 (1.7) | |
| 55–64 years | 13195 (23.6) | 180 (6.1) | |
| >65 years | 34780 (62.2) | 2700 (92.2) | |
| **LOS group (%)** | | | <0.001 |
| 1–9 | 42585 (71.1) | 1950 (73.0) | |
| 10–19 | 7160 (11.9) | 435 (16.3) | |
| 20–29 | 5180 (8.6) | 110 (4.1) | |
| >30 | 5005 (8.4) | 175 (6.6) | |

Abbreviations: SD, Standard deviation; NCHS, National Center for Health Statistics; $, United States' Dollar. LOS: Length of Stay.

Note: All frequencies and percentages are weighted

Research indicate that thirty to fifty percent of cancer patients will have a mental health condition during their cancer journey, according to the evidence that is currently available. Depression, anxiety, and trauma-related disorders are all part of this group [38]. An additional 15–20% of cancer patients also suffer from clinically significant pain, health anxiety, a lack of meaning in life, and other existential issues that are not classified as mental disorders [39]. The prevalence of major depressive disorders in cancer patients is three times higher than the general population, according to estimates [15–29, 40, 41]. In the cancer community, 8–24% of patients report depressive symptoms, and additionally, mental health issues are more common among cancer patients in advanced stages and those receiving palliative care [42, 43]. A high prevalence of mental health difficulties is associated with the emotional and psychological toll that cancer takes on individuals. During the cancer diagnosis, treatment, and aftercare processes, survivors often feel anxious, depressed, and distressed [44]. Since the symptoms of cancer and mental health issues frequently overlap, it can be challenging to detect both and management requires diagnosis and recognition. Evidence suggests that cancer patients with co-occurring mental health disorders are less likely to adhere to their treatment plans, which may result in worse health outcomes [45]. Screening for and providing support for mental health issues is essential to improving the overall health and prognosis of cancer patients. Furthermore, our research reveals that cancer patients who also have MI are more likely to experience septicemia, weight loss, and fluid and electrolyte imbalance. This implies the incremental costs burden for cancer patients with MI. Due to the prevalence of mental health issues in cancer patients, the impact of these issues can vary depending on their health-seeking behaviors, and sociodemographic factors. As the results remained consistent across all cancer cohorts and the sub-groups, the findings implies the need to screen cancer patients for their met and unmet needs in all stages of their cancer.

The prevalence of cancer is increasing worldwide at an ever-increasing rate. Survivorship in cancer care and mental health care require attention on mental illness. The impact of cancer upon mental health and impact of management of cancer and MI impact the cost of care in hospitalized cancer patients. It is anticipated that almost half of cancer patients may experience some form of MI during their journey with the disease [38]. Additionally, cancer diagnosis has significant psychosocial consequences for patients and caregivers; when appropriate interventions were applied, it culminated in better results, as shown by extensive research conducted over the past few decades [27, 46–48]. However, the magnitude of the outcomes are multiple and complex [49], including economic and psychosocial. Recent reports also indicate that mental illnesses are associated with the increased risk of infectious complications following cancer diagnosis [50]. Nonetheless, a patient's health status cannot be accurately measured using a single primary outcome due to the complexity of several mental illnesses and their impact [51]; in addition, patient-defined outcomes, intermediate outcomes, adverse outcomes, and financial consequences are not adequately evaluated most of the time [52]. The research undertaken by Liu, et al. [51] investigated the psychological disorders and consequences of septicemia. Furthermore, Lui et al examined the occurrence of pre-cancer mental illness. Unfortunately, we were unable to evaluate this variable due to limitations in our dataset. The results in the hematological patient's research differ from ours primarily because our cohort included a large number of patients whereas the Liu, et al, study only included a limited number of individuals who may have received prophylactic antibiotics.

It is insufficient to evaluate MI solely on one result as opposed to a battery of outcomes. As Grassi, et al captured existential conditions such as demoralization, health anxiety, hopelessness, and existential distress in cancer patients across stages and types of cancer treatment, [53] and further discussed by Caruso, et al, these conditions are not included as 'disorders' in conventional diagnostic and nosological systems [54]. While we explored costs associated as

an outcome, consequently, an economic evaluation to detect the effects of MI may provide a limited overall estimate. However, the results may be affected by a small proportion, nevertheless, the outcomes could be influenced by a minority group that may depend on healthcare accessibility, racial disparities, and equitable distribution of health resources [55]. For the last two decades, Quality of Life assessment has been widely used along quantitative measures, Quality-adjusted life-year (QALY) gains––cost per QALY gained––are a metric used in economic evaluations to quantify the health benefit measure utilized to determine the cost of treatments [56].

It should be pointed out that past studies show a 30% economic gap between cancer patients with poor mental health and those with good mental health [57, 58]. In our study, we found that LC and OPC with MI had higher total charges in this study, indicating that the LC and OPCs have a higher associated complication. In addition, in all cancer groups, patients with MI had longer LOS. It is important to note that in our study, there is a higher number of patients with MI who transferred out to different facilities, and thus the actual cost may be different. According to the most recent Annual Report to the Nation on the Status of Cancer, Part 2, cancer patients in the United States pay a disproportionately high share of cost for cancer care. The national patient economic burden for cancer care in 2019 was $21.09 billion, including $16.22 billion in patient out-of-pocket costs and $4.87 billion in patient time costs [59]. Studies reveal that patients with post-cancer mental diseases suffered the highest expenditures; the greater costs incurred by cancer patients who were later determined to have concurrent mental conditions could be attributed to additional office visits for mental health care as well as an increase in the amount of medication they used [60]. Unfortunately, our data set did not allow examination of the post-cancer treatment options for these patients.

Although infectious complications and MI are most common among cancer patients, the association between the two is not well understood. Psychological stress, which is recognized to play a major role in the onset, maintenance, and aggravation of mental disorders–as well as in modifying immunological function–is a possible pathway linking MI with infections [61, 62]. In this regard, our study showed that among all cancer cohorts, MI were associated with the septicemia. This is an area which needs more attention and a more serious implication. Liu et al. [51] reported that a statistically significant association between precancer psychiatric disorders and sepsis present between both of their simplified hazard ratios (HR), 1.31; 95% CI, 1.22–1.40, and full HR, 1.26; 95% CI, 1.18–1.35. Additionally noting the findings from a meta-analysis, a pooled relative risk of 2.22 (95% CI, 2.12–2.33) of mortality among people with mental illnesses [62]. Other researchers have also shown a strong association of MI and mortality [63–65]. Additionally, our research shows that even after controlling for covariates such as co-morbidities (Elixir Score) in the multivariate analysis, cognitive problems were linked to in-hospital mortality among PC patients. Multiple studies have linked different degrees of cognitive impairment to an increased risk of death [66, 67]. Therefore, it is possible that functional deterioration and underlying co-morbidities are the significant risk factors linked to the aspect's mortality risk.

Nonetheless numerous risk factors have been linked to MI––old age, poor health, and obesity have been identified as major contributors [66–70]. While it is acknowledged that MI are multifactorial maladies caused by the interaction of environmental factors and genetic predisposition, their impact on the aging population with multiple diseases, including cancer, is much greater [71, 72]. In our study, MI were associated with fluid and electrolyte imbalance, and weight loss. Reports indicate that people with MI, especially those in the early stages were associated with weight loss, which was linked to a decrease in food consumption, and malnutrition [73–75]. On the contrary, being overweight reduced the risk of cognitive disorder related mortality [76]. In other words, the health outcomes of patients with MI appear to be

poorer for those who are underweight. However, it is not advisable to be either overweight or underweight due to the potential negative health implications.

Despite completing their treatment, numerous cancer survivors still grapple with significant mental health issues [38]. Cancer survivors have a greater incidence of mental health disorders, such as depression, anxiety, and psychotic disorders, in comparison to the general population. Individuals who have survived cancer throughout are at a significantly increased risk of getting major depressive disorder, with a fivefold higher likelihood compared to their healthy counterparts [77]. Individuals who have survived colorectal and breast cancer have a higher likelihood, twice as much, of experiencing difficulties with concentration and memory compared to those who have survived other types of cancer [78, 79]. Female cancer survivors more frequently experience cognitive impairments compared to their male counterparts [80]. Further, lower income, social support, and education level are characteristics associated with a higher prevalence of mental health issues among cancer survivors [81]. Enduring cancer can be distressing and disruptive, exacerbating psychological distress. While it is common for cancer survivors to feel anxious and unhappy after treatment, it is important to note that many survivors may still face ongoing mental health challenges [82]. Continuous assistance and supervision are necessary. In order to provide comprehensive care, it is essential to address the mental health concerns of those who have survived cancer. Clinicians should routinely assess survivors for mental health challenges such as anxiety, depression, and cognitive decline, and thereafter connect them with appropriate support services to aid in their coping.

Patients who are diagnosed with cancer are more likely to suffer from psychiatric disorders, and the cost of cancer that is accompanied by a mental problem is not proportional to the total cost of both conditions [83]. The research that investigates the connection between cancer and MI is still in its infancy, despite the fact that the topic is becoming increasingly important. This is due to the fact that cancer and mental disease are given a disproportionate amount of weight in the profile of healthcare expenditures.

## Limitations

These limitations must be considered when evaluating the findings of the study. First, as with any large database, the quality of data input is primarily determined by those providing the information and is susceptible to variations in reporting and coding. Nonetheless, inconsistencies are believed to be significantly mitigated by the quantity of data included. Secondly, since NIS data includes discharges, it is theoretically possible for a patient to be readmitted and counted more than once, despite the fact that this is exceedingly unlikely, and there are no data on the population of non-admitted patients. Finally, when evaluating resource utilization, we were limited to hospital charges and not actual costs. Even though we controlled potential confounding factors in the study, it is conceivable that there are variables associated with exposure status that we were unable to include in our investigation due to limitations imposed by the dataset. In addition, our analyses only include cancer patients who are hospitalized; hence, the findings may not be applicable to a wider population.

## Conclusion

Cancer patients have a higher risk of having pre-existing mental disorders or developing mental illness, which is more debilitating during cancer recovery––all cancer cohorts in the study demonstrated a significant impact of MI, on the burden of illness, weight loss, fluid and electrolyte imbalance, and septicemia. Because of its profound impact on both patients and caregivers, ongoing longitudinal study into cancer survivors' mental health is crucial.

## Supporting information

**S1 File. ICD 10 CM billable codes.**
(DOCX)

## Acknowledgments

We thank Ishan S for organizing the tables, and Vipanchika S for the organizing the tables grammatical corrections.

## Author Contributions

**Conceptualization:** Poolakkad S. Satheeshkumar, Joel B. Epstein.

**Data curation:** Poolakkad S. Satheeshkumar.

**Formal analysis:** Poolakkad S. Satheeshkumar.

**Investigation:** Poolakkad S. Satheeshkumar.

**Methodology:** Poolakkad S. Satheeshkumar.

**Project administration:** Poolakkad S. Satheeshkumar.

**Resources:** Poolakkad S. Satheeshkumar.

**Software:** Poolakkad S. Satheeshkumar.

**Supervision:** Poolakkad S. Satheeshkumar, Roberto Pili, Joel B. Epstein.

**Validation:** Poolakkad S. Satheeshkumar.

**Visualization:** Poolakkad S. Satheeshkumar.

**Writing – original draft:** Poolakkad S. Satheeshkumar.

**Writing – review & editing:** Poolakkad S. Satheeshkumar, Roberto Pili, Sudheer B. Kurunthatil Thazhe, Rhine Sukumar, Minu Ponnamma Mohan, Eric Adjei Boakye, Joel B. Epstein.

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
