## [Decision Letter · Decision Letter 0]

29 Feb 2024

PMEN-D-24-00014

Impact of Cognitive Disorders On Burden of Illness, Septicemia and Mortality Among Patients Hospitalized for Cancer.

PLOS Mental Health

Dear Dr. Poolakkad, 

Thank you for submitting your manuscript to PLOS Mental Health. After careful consideration, we feel that it has merit but does not fully meet PLOS Mental Health’s publication criteria as it currently stands. Therefore, we invite you to submit a revised version of the manuscript that addresses the points raised during the review process.

Please submit your revised manuscript by **March 28, 2024.** If you will need more time than this to complete your revisions, please reply to this message or contact the journal office at mentalhealth@plos.org. Please include the following items when submitting your revised manuscript:

We look forward to receiving your revised manuscript.

Kind regards,

Kizito Omona, PhD

Academic Editor

PLOS Mental Health

Journal Requirements:

1. Please amend your online Financial Disclosure statement. If you did not receive any funding for this study, please simply state: “The authors received no specific funding for this work.”

2. Please update your online Competing Interests statement. If you have no competing interests to declare, please state: “The authors have declared that no competing interests exist.”

Additional Editor Comments (if provided):

I can see that the paper is fairly well written and is insightful. However, I believe that it can still be made much better. Critically address the comments of the reviewers to improve your paper.

Reviewers' comments:

Reviewer's Responses to Questions

**Comments to the Author**

1. Does this manuscript meet PLOS Mental Health’s publication criteria? Is the manuscript technically sound, and do the data support the conclusions? The manuscript must describe methodologically and ethically rigorous research with conclusions that are appropriately drawn based on the data presented.

Reviewer #1: Yes

Reviewer #2: Yes

2. Has the statistical analysis been performed appropriately and rigorously?

Reviewer #1: Yes

Reviewer #2: Yes

3. Have the authors made all data underlying the findings in their manuscript fully available (please refer to the Data Availability Statement at the start of the manuscript PDF file)?

Reviewer #1: Yes

Reviewer #2: Yes

4. Is the manuscript presented in an intelligible fashion and written in standard English?

Reviewer #1: Yes

Reviewer #2: Yes

5. Review Comments to the Author

Reviewer #1: An interesting and trendy research that addresses a very relevant issue with clear results and methods, As described by the authors the association of patients with cancer and higher risk of developing mental illness and its impact.

Reviewer #2: Thank you for submitting your work to PMEN journal.

Research submitted to this publication regarding the impact of cognitive disorders on the burden of illness, septicemia, and mortality among cancer patients hospitalized may not have conducted a thorough literature analysis or developed hypotheses for a number of reasons.

To start with, there could not be any recent or relevant studies to back up the authors' study topic because they didn't do a thorough search of the literature. If you want to know what others know about a subject right now, you need to do a literature search. This means looking through a lot of databases, making sure you use the right keywords, and incorporating both published and unpublished studies.

Second, it's possible that the writers didn't do a thorough job of evaluating the research that were part of their literature review. Research studies must undergo critical appraisal to guarantee their results are solid and relevant to the study's topic by checking their validity, reliability, and generalizability.

Thirdly, after reviewing the literature, the writers might not have come up with solid, testable theories. If you want your research to go in the right path, you need to formulate specific, quantifiable hypotheses based on what you found in the literature review.

Finally, the research aims and procedures can be murky because the writers didn't adequately justify their study's design and research question. A well-reasoned rationale will clarify the significance of the research question, identify the knowledge gap that the study intends to fill, and explain the rationale behind the selection of the study design to answer the research question.

The authors could address these potential issues by performing a thorough literature search using appropriate search terms across multiple databases, critically evaluating the studies they include, using the literature review to develop clear and testable hypotheses, and clearly justifying their research question and study design. The writers can improve the research's rigor and impact by fixing these problems with the literature evaluation and hypothesis creation.

6. PLOS authors have the option to publish the peer review history of their article (what does this mean?). If published, this will include your full peer review and any attached files.

**Do you want your identity to be public for this peer review?** For information about this choice, including consent withdrawal, please see our Privacy Policy.

Reviewer #1: **Yes**

Reviewer #2: No

---

## [Decision Letter · Decision Letter 1]

27 Aug 2024

PMEN-D-24-00014R1

Impact of Cognitive Disorders On Burden of Illness, Septicemia and Mortality Among Patients Hospitalized for Cancer.

PLOS Mental Health

**Dear Dr. Satheeshkumar,**

Thank you for submitting your manuscript to PLOS Mental Health. I appreciate the improvement made. However, after careful consideration, we feel that it has merit but does not yet fully meet PLOS Mental Health’s publication criteria as it currently stands. Therefore, we invite you to submit a revised version of the manuscript that addresses the points raised during the review process.

EDITOR:

Your work has a lot of merits. You need to spend a little more time refine it as raised by the reviewers.

Please submit your revised manuscript by **26th September 2024.** If you will need more time than this to complete your revisions, please reply to this message or contact the journal office at mentalhealth@plos.org. Please include the following items when submitting your revised manuscript:

We look forward to receiving your revised manuscript.

Kind regards,

Kizito Omona, PhD

Academic Editor

PLOS Mental Health

Journal Requirements:

Additional Editor Comments (if provided):

Carefully address the comments raised by the reviewers as fast as you can. Your work has a lot of merit but there is need for some few refinement as noted by the reviewers.

Reviewers' comments:

Reviewer's Responses to Questions

**Comments to the Author**

1. If the authors have adequately addressed your comments raised in a previous round of review and you feel that this manuscript is now acceptable for publication, you may indicate that here to bypass the “Comments to the Author” section, enter your conflict of interest statement in the “Confidential to Editor” section, and submit your "Accept" recommendation.

Reviewer #2: All comments have been addressed

Reviewer #3: All comments have been addressed

Reviewer #4: (No Response)

2. Does this manuscript meet PLOS Mental Health’s publication criteria? Is the manuscript technically sound, and do the data support the conclusions? The manuscript must describe methodologically and ethically rigorous research with conclusions that are appropriately drawn based on the data presented.

Reviewer #2: Yes

Reviewer #3: Yes

Reviewer #4: Partly

3. Has the statistical analysis been performed appropriately and rigorously?

Reviewer #2: Yes

Reviewer #3: Yes

Reviewer #4: Yes

4. Have the authors made all data underlying the findings in their manuscript fully available (please refer to the Data Availability Statement at the start of the manuscript PDF file)?

Reviewer #2: Yes

Reviewer #3: Yes

Reviewer #4: Yes

5. Is the manuscript presented in an intelligible fashion and written in standard English?

Reviewer #2: Yes

Reviewer #3: Yes

Reviewer #4: No

6. Review Comments to the Author

Reviewer #2: Good luck

Reviewer #3: The study, "Impact of Cognitive Disorders on Burden of Illness, Septicemia, and Mortality Among Patients Hospitalized for Cancer," presents a well-researched and timely analysis of a critical issue in cancer care. By addressing the intersection of cognitive disorders, septicemia, and mortality, the authors highlight important clinical implications and underscore the need for holistic care approaches for cancer patients. The methodology was robust, and the findings contribute significantly to the literature, offering valuable insights that could inform future healthcare strategies and interventions.

Recommendation:

While the introduction touches on important topics, it lacks clarity in establishing the connection between cognitive disorders, septicemia, and mortality in cancer patients. The rationale behind the study could be further elaborated, as the current narrative does not sufficiently explain the gap in the literature that the research intends to fill. Additionally, the introduction could benefit from a more thorough review of relevant existing studies to better contextualize the study's aims. This would help provide a stronger foundation for the research.

Reviewer #4: The manuscript needs to be revised well. I have indicated this specifically in my attached comments. Special attention should be given for the introduction, results, and discussion and the overall write up of the manuscript. With adequate revisions, i hope that the study will be of great value.

7. PLOS authors have the option to publish the peer review history of their article (what does this mean?). If published, this will include your full peer review and any attached files.

**Do you want your identity to be public for this peer review?** For information about this choice, including consent withdrawal, please see our Privacy Policy.

Reviewer #2: No

Reviewer #3: **Yes: **Jesan Ara

Reviewer #4: No

---

## [Editor Report · Decision Letter 2]

13 Sep 2024

Impact of Cognitive Disorders On Burden of Illness, Septicemia and Mortality Among Patients Hospitalized for Cancer.

PMEN-D-24-00014R2

Dear Dr. Satheeshkumar,

We are pleased to inform you that your manuscript 'Impact of Cognitive Disorders On Burden of Illness, Septicemia and Mortality Among Patients Hospitalized for Cancer.' has been provisionally accepted for publication in PLOS Mental Health.

Best regards,

Kizito Omona, PhD

Academic Editor

PLOS Mental Health